# MSmart: Smart Contract Vulnerability Analysis and Improved Strategies Based on Smartcheck

**Jiajia Fei, Xiaohan Chen and Xiangfu Zhao ***

School of Computer and Control Engineering, Yantai University, Yantai 264005, China
* Correspondence: xiangfuzhao@ytu.edu.cn

**Abstract:** As is well known, smart contracts on the blockchain store plenty of digital assets, and these contracts deployed on the blockchain are difficult to be modified. For this reason, the analysis and detection of smart contract vulnerabilities have received extensive attention. Smartcheck, a typical Java-implemented static analysis tool of smart contracts, is capable of converting Solidity source code into path diagrams based on the lexical and syntactic analysis, and finds smart contract vulnerabilities by path matching. Although Smartcheck can analyze most of the real-world vulnerabilities, some imperceptible vulnerabilities may be ignored, causing huge economic losses. In order to address these issues, we develop a new tool named MSmart to analyze the vulnerabilities of high risk such as timestamp dependence vulnerabilities, integer overflow vulnerabilities, self-destruct vulnerabilities, etc. MSmart converts the smart contract source code into an intermediate representation, and looks for smart contract vulnerabilities based on intermediate representation and XPath rules. We add new intermediate representation rules of Smartcheck to detect more kinds of vulnerabilities and optimize existing rules to suit the complexity of smart contract. We also implemented smart contract batch detection to shorten the time it takes to find vulnerabilities. To analysis the performance of MSmart, we collect 6000 real-world contracts from Etherscan and design some comparative experiments with other tools. The results of experiment show that MSmart is able to analyze related vulnerabilities better, and false positives and false negatives have been reduced due to our improvements.

**Keywords:** smart contract; security vulnerability; Smartcheck; blockchain

## 1. Introduction

Blockchain is the core technology of Bitcoin. It first appeared in "Bitcoin: A Peer-to-Peer Electronic Cash System" [1] published by Satoshi Nakamoto. The essay narrates a new, decentralized, completely distributed payment system architecture that does not require a trust foundation. One prominent use of blockchain technology is the smart contract. As early as the 1990s, the concept of smart contracts was proposed by Nick Szabo [2]. Before the advent of blockchain technology, smart contracts were not commonly used for technical reasons [3]. Smart contracts are automatically triggered when preset circumstances are satisfied, unlike traditional contracts which require a reliable third party [4]. Tens of thousands of smart contracts have been deployed on blockchain platforms and are continuing to grow rapidly. However, as the quantity of smart contracts rises, there are already significant challenges with their security. Smart contracts cannot be modified once they have been deployed because of their immutability and irreversibility. The deployment of smart contracts is in an open network setting. Therefore, smart contracts are more easily attacked [5,6]. Anonymous attackers can obtain higher remuneration, easier cash outs (ether can be traded immediately), and a lower risk of being punished by attacking smart contracts, which is more likely to result in victims taking huge financial losses. In 2017, the multi-signature wallet Parity lost hundreds and thousands of ether, and the delegatecall vulnerability caused millions of funds to be frozen. In 2018, USChain BEC lost about 900 million dollars due to the integer overflow vulnerability. In 2020, many

hackers attacked smart contract games including FarmEOS, Playgames, LuckBet, and EOSPlaystation [7–10], which lost a total of around one million dollars. These show that the smart contract vulnerability has caused the significant financial losses and the significance of smart contract security. Therefore, before being implemented, smart contracts must be checked to make sure they are accurate [11].

In order to ensure property safety, a tool that can detect smart contract vulnerabilities is essential. The current smart contract vulnerability detection technologies mainly include symbolic execution, intermediate representation, etc. Oyente [12] uses symbolic execution to detect smart contract vulnerabilities. Oyente has significantly reduced the false positive by traversing each path and using the Z3 solver [13]. Oyente was successful in limiting the number of cycles to prevent the space explosion [14], which also leads to a significant increase in the false negative rate. Another tool for symbolic execution to find vulnerabilities is Mythril [15], and it detects smart contract vulnerabilities by using conceptual analysis [16], taint analysis [17], and verification control flow [18]. While Mythril can analyze many types of smart contract vulnerabilities, it needs to find critical bytecodes (such as "ADD"), which takes a long time.

Slither [19] uses the intermediate representation to detect smart contracts vulnerabilities. Slither offers syntax checking in addition to supporting contract checking. Slither is prominent in reentrant vulnerability detection. Another tool for the intermediate representation to find vulnerabilities is Smartcheck [20]. Smartcheck is a scalable static smart contract analysis tool. Although Smartcheck can analyze most of the existing smart contract vulnerabilities, it also has some false positives and false negatives. In order to better identify common vulnerabilities such as timestamp dependence vulnerabilities, integer overflow vulnerabilities, self-destruct vulnerabilities, etc., we improve Smartcheck. The improved Smartcheck, named MSmart, has carried out detailed comparison experiments on real large data sets. Compared with other tools such as Oyente via specific vulnerability contracts, it is concluded that MSmart can better analyze and detect related vulnerabilities. At the same time, the false positives and false negatives of vulnerabilities have been reduced, thus providing higher security for smart contracts.

## 2. Preliminary Knowledge

### 2.1. Smart Contracts and Vulnerabilities

#### 2.1.1. Integer Overflow

When the size of the value exceeds the top or lower constraint for that data type, there is an integer overflow vulnerability [21]. Integer overflow is generally divided into overflow and underflow. Addition overflows, multiplication overflows, and division overflows are the three most frequent overflows in smart contracts. An integer overflow vulnerability occurs in smart contracts when the range of values of an integer variable is exceeded by the arithmetic operation. The unsigned integer uint8 stores the range [0, 255], for instance, (uint8) 255 + 1 will be incorrectly stored as 0. Similarly, (uint8) 0 − 1 will be incorrectly stored as 255.

Figure 1 shows a section of code from the USChain BEC contract [22] that has an integer overflow. The entry "uint amount= uint256 (cnt) * _value" at line 6 has an integer overflow vulnerability. The data types of "cnt", "_value," and "amount" are uint256; if the incoming value is too high and "amount" exceeds the maximum value, it will be mistakenly saved as 0, since the storage range of uint256 is [0, 2^256 − 1]. The contract owner lost a significant amount of ether as a result of this vulnerability.

#### 2.1.2. Timestamp Dependence

When smart contract execution is dependent on block timestamps, timestamp dependence [23,24] vulnerabilities might arise. The results of the prior run will also change if the block timestamps are different. The block timestamp in the contract is determined by the miner's local time. Theoretically, the local time might be changed by miners. Because of this, using the block timestamp as a pseudo-random number in the contract to

carry out certain important operations (such as sending ether) may trigger timestamp dependence vulnerabilities.

```
1   contract PausbaleToken{
2      ......
3      function batch Transfer(address[] _receivers,uint256 _value)
4      public whenNotPaused returns(bool){
5          uint cnt = _receivers.length;
6          uint256 amount = uint256(cnt) * _value ;
7          require(cnt > 0 && cnt < =20);
8          require(_value > 0 && balances[msg.sender] >= amount );
9          ......
10          return true;
11     }
12  }
```

**Figure 1.** Integer overflow vulnerability case.

An example of a timestamp dependence vulnerability is shown in Figure 2. The gambling contract Roulette uses the block timestamp to decide who can obtain all the bets. Line 5 of the contract determines the stake is 10 ether. The winning condition is established in lines 8–9 (now% 5 == 0), and the winner will obtain all the bets. The miner has prior knowledge of whether the win condition will be met since the block timestamp is generated by the miner. This vulnerability can be found after Smartcheck analysis. However, Smartcheck generates a lot of false negatives while examining the following timestamp dependence vulnerabilities.

```
1   contract Roulette {
2     uint public pastBlockTime;
3       constructor() public payable{}
4       function() public payable{
5         require(msg.value == 10 ether);
6         require(now! = pastBlockTime);
7         pastBlockTime = now;
8         if ( now % 5 == 0) {
9           msg.sender.transfer(this.balance);
10        }
11     }
12  }
```

**Figure 2.** Timestamp dependence code of contract Roulette.

The TimeFame1 contract in Figure 3 is a block timestamp-related instance. It uses block timestamps to decide who takes the balance in the contract. Line 5 confirms that the stake to participate is 1 ether. Who receives the balance is decided by the win condition (lastBlockTime % 15 == 0) at line 8. In this contract, the block timestamp (block.timestamp) is assigned to lastBlockTime. Therefore, Smartcheck doesn't report it as a vulnerability.

```
1   contract TimeFame1{
2     uint public lastBlockTime;
3     function lucky() public payable {
4       require (msg.value == 1 ether );
5       require (lastBlockTime! = block.timestamp);
6       lastBlockTime = block.timestamp;
7       if (lastBlockTime % 15 == 0){
8         msg.sender.transfer(address(this).balance);
9       }
10    }
11  }
```

**Figure 3.** Timestamp dependence code of contract TimeFame1.

### 2.1.3. Self-Destruct

The self-destruct [25] function can transfer the ether to the target address. Once ether is involved, the attacker is likely to alter the address in order to transfer the ether from the contract to his own account. The self-destruct function allows the attacker to transfer ether to the target contract for his own goal. At the same time, the use of the self-destruct function will affect the contract status. Tens of millions of dollars were lost as a result of the parity multi-signature wallet vulnerability [26] in 2017 due to this vulnerability.

We can observe Figure 4 to find: each time, the player sends 1 ether to the EtherGame contract, which is checked to see whether the balance is less than 8, and only then is the function able to proceed. The contract will collapse if an attacker uses the self-destruct function to force ether to be transferred into the EtherGame.

```
1   pragma solidity ^0.8.10;
2   contract EtherGame {
3     uint public targetAmount = 7 ether;
4     address public winner;
5     function deposit() public payable {
6       require(msg.value == 1 ether, "You can only send 1 Ether");
7       uint balance = address(this).balance;
8       require(balance <= targetAmount, "Game is over");
9       if (balance == targetAmount) {
10        winner = msg.sender;
11      }
12    }
13    function claimReward() public {
14      require(msg.sender == winner, "Not winner");
15      (bool sent, ) = msg.sender.call{value: address(this).balance}("");
16      require(sent, "Failed to send Ether");
17    }
18  }
```

**Figure 4.** Self-destruct vulnerability code.

### 2.1.4. Delegatecall

The so-called delegatecall [27,28] means a proxy call. This means that when the delegatecall function is called, the value of the built-in variable "message" will vary as the state does. The values of "*msg.sender*" and "*msg.value*" remain unchanged when utilizing the delegatecall function, but the execution environment is altered to that of the caller's (proxy) operating environment. As a result, calling a smart contract defined by a "stateful" library will have an indirect impact on the current smart contract's state.

The wallet contract is shown in Figure 5, it can be found that if *msg.data.length* > 0, the function call of delegatecall will be triggered. It is observed that the wallet library function can be self-destructed. If the self-destruct function is called, the state will change. This will cause the fallback function of the wallet contract to permanently return 0 (the target address has no associated code), and the balance in it will be permanently locked and no further operations can be performed.

### 2.1.5. Denial of Service

Denial of service (also known as DOS) [29–31] describes the excessive use of contract resources (such as ether or gas) to prevent the contract from achieving the anticipated execution purpose. The main common attack methods are as follows: (1) The smart contract will not be carried out when it's state changes and certain predetermined criteria are not satisfied. This may result in a DOS attack. (2) The attacker makes the gas consumption in the contract exceed the gas limit, and (3) the smart contract owner's account is not properly protected. Since transactions may be opened and closed by the owner's account, the transaction is likely to be frozen if the attacker gains ownership of the smart contract.

```
1   contract WalletLibrary is WalletEvents {
2     ······
3     function kill(address _to) onlymanyowners(sha3(msg.data)) external {
4       suicide(_to);
5     }
6     ······
7   }
8   contract Wallet is WalletEvents {
9     ······
10    function() payable {
11      if(msg.value > 0)
12        Deposit(msg.sender, msg.value);
13      else if (msg.data.length > 0)
14        WalletLibrary.delegatecall( msg.data);
15    }
16  }
```

**Figure 5.** A typical example of delegatecall vulnerability.

Figure 6 shows a simplified version of the KotEt contract [32]. The main reason for vulnerabilities in this contract is in lines 5-6. First, the fifth line is carried out. If a new bidder offers a higher price than the previous lead bidder, the contract will refund the amount paid by the previous lead bidder for the bid. The fallback function is triggered at this time, but if there is an unrealizable function in the attacker's fallback function. Therefore, the DOS attack is triggered when the contract becomes stuck returning the bid amount.

```
1   contract auction {
2     address public currentleader;
3     uint public highestbid;
4     function bid() public payable {
5       require( msg.value > highestbid);
6       require( currentleader.sender.send(highestbid));
7       currentleader = msg.sender;
8       highestbid = currentleader ;
9     }
10  }
```

**Figure 6.** A simplified version of the KotEt contract.

*2.2. Symbolic Execution and Intermediate Representation*

The main methods of analysis of smart contracts are static analysis and dynamic detection. Static analysis mainly includes intermediary representation method, symbolic execution method, and so on. The symbolic execution and intermediate representation will next be briefly discussed.

The intermediary representation method converts the source code or bytecode of a smart contract into an intermediary representation (IR) with high semantic expression, and then analyzes the intermediary representation of the contract to find vulnerabilities. The symbolic execution approach may also be used to find smart contract vulnerabilities. The constraint solver is used to solve the constraint and determine the input to the execution, and finally the constraint solver is used to obtain new test inputs and detect the presence of vulnerabilities. Smartcheck is a typical static analysis tool.

*2.3. Smartcheck's Detection Method*

Of the five vulnerabilities, Smartcheck can detect timestamp dependence vulnerabilities and self-destruct vulnerabilities.

For timestamp dependence: Smartcheck detects the presence of "==" after the block timestamp (such as "now % 5 == 0"). With this kind of detection, it is clear that there are a lot of false negatives.

For self-destruct: Smartcheck detects the presence of the suicide function. With this kind of detection, it is clear that there are a lot of false positives.

## 3. MSmart

### 3.1. MSmart

Smartcheck is a static analysis tool for smart contracts implemented in Java. MSmart follows Smartcheck's detection framework. It uses the lexical and syntactic analysis to convert Solidity [33] source code into path diagrams, as shown in Figure 7. MSmart uses the ANTLR [34] language parser and Solidity grammar to produce an XML parse tree as an intermediate representation. Finally, MSmart identifies the vulnerability through XML Path Language (referred to as Xpath, which is a language for finding information in XML documents) rule matching [35].

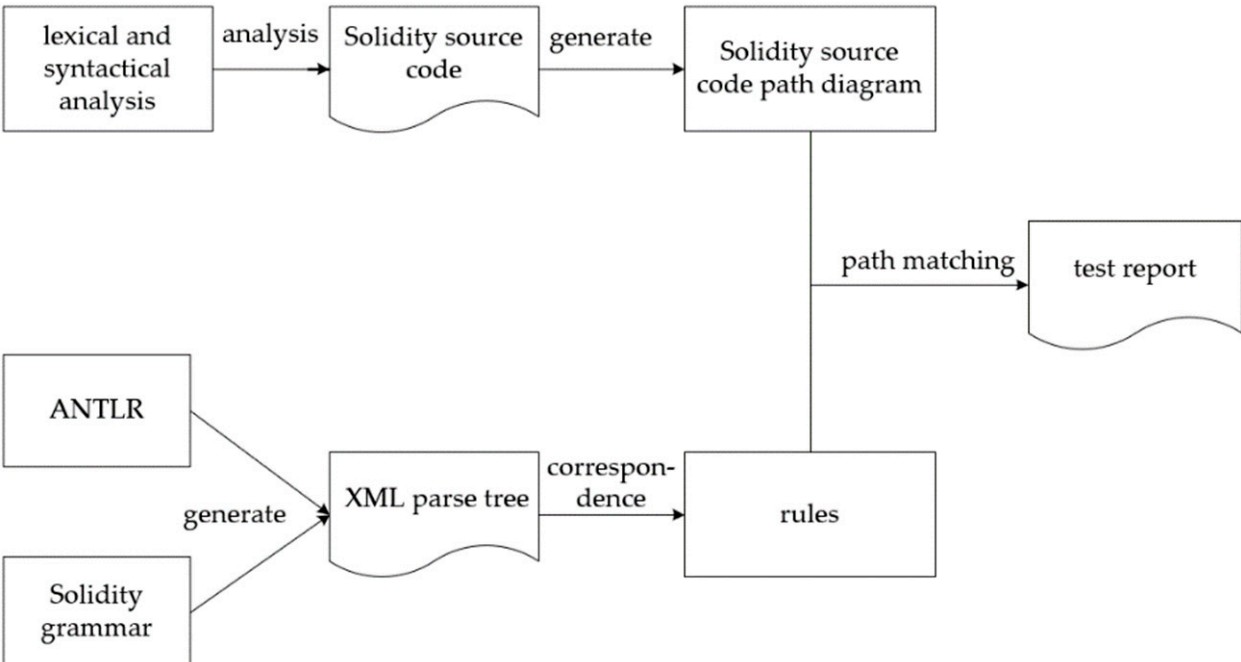

**Figure 7.** MSmart analysis flow chart.

### 3.2. MSmart's Batch Test Function

The time it takes to test large data sets is significantly reduced by MSmart's addition of a batch testing feature to the original Smartcheck. We created a Ttest class, which is shown in Figure 8, that just takes the location to the folder containing the smart contracts as input, and produces a.txt or.csv file as output. This simplifies the manual tagging process for smart contracts.

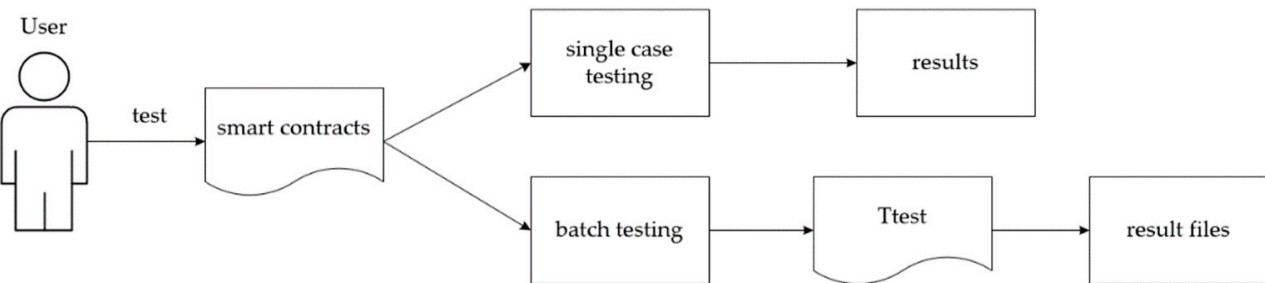

**Figure 8.** Introduction of Ttest.

### 3.3. XML Parse Tree

The XML parse tree of the contract code (uint256 (cnt) * _value) is shown in Figure 3: we will choose an example to demonstrate how MSmart transforms Solidity source code into the appropriate path diagram.

Figure 9 shows the tree structure of the (uint256 (cnt) * _value) code expression, in which the root node is "expression", each child node is "expression","muldivOperator" and "expression", and the leaf nodes are "uint256", "cnt", "*", and "_value". To illustrate how to find the "uint256 (cnt) * _value" by path matching, the "expression" of the root node is first determined and then find the leaf nodes "*" and "uint256".

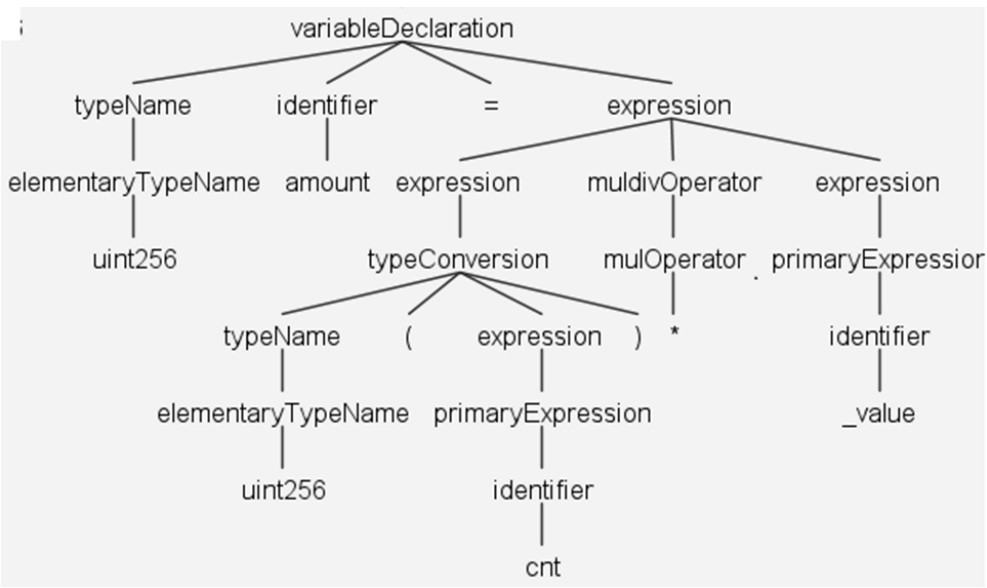

**Figure 9.** uint256 (cnt) * _value code analysis tree.

### 3.4. Differences between MSmart and Smartcheck

Our summary is shown in Table 1 to help readers better understand the similarities and differences between MSmart and Smartcheck:

**Table 1.** Differences between MSmart and Smartcheck.

| Differences | MSmart | Smartcheck |
| --- | --- | --- |
| integer overflow | ✓ | ✗ |
| timestamp dependence | ✓ | ✓ |
| self-destruct | ✓ | ✓ |
| delegatecall | ✓ | ✗ |
| DOS | ✓ | ✗ |
| batch detection | ✓ | ✗ |

MSmart adds detection for integer overflow vulnerabilities, delegatecall vulnerabilities and DOS vulnerabilities, and MSmart also supports batch detection. Both MSmart and Smartcheck can detect for timestamp dependence vulnerabilities and self-destruction vulnerabilities, but MSmart adds some rules to reduce false positives and false negatives.

## 4. MSmart's Detection Method

### 4.1. Integer Overflow

The integer overflow vulnerability satisfies two basic conditions: (1) an arithmetic operation occurs, and (2) the arithmetic operation has an uint data type on one side. If both conditions are met and there is no integer overflow protection, it can be concluded that there is an integer overflow vulnerability. In the experiment, the integer overflow protection

is taken into account in order to reduce false positives. Integer overflow vulnerabilities may presently be protected against in two ways: (1) by using the standard SafeMath [36] library, and (2) by using the "if" or "require" judgment statements. To guard against integer overflow vulnerabilities, for instance, use the line "if (uint256 (cnt) * _value 2^256 - 1)" as a judgment statement.

Based on the above analysis, the basic ideas for improvement are shown in Algorithm 1:

The integer overflow detection algorithm is shown in Algorithm 1.

---

**Algorithm 1:** The integer overflow detection algorithm.

---

**Input:** a smart contract P
**Output:** integer overflow vulnerabilities
1: Convert the source code to the corresponding path diagram
2: Match according to the path expression pattern
3: **if** pattern matching found arithmetic operation **then**
4:     **if** the data type involved in the arithmetic operation is the uint type **then**
5:         **if** not using the standard library function SafeMath **then**
6:             **if** without using if or require statement guards **then**
7:                 **return** integer overflow vulnerability
8:             **else**
9:                 **return** 0
10:             **end if**
11:         **else**
12:             **return** 0
13:         **end if**
14:     **else**
15:         **return** 0
16:     **end if**
17: **else**
18:     **return** 0
19: **end if**
20: Integrate all discovered vulnerabilities

---

In the integer overflow detection algorithm, we mainly check the contract in the following way: (1) The data are examined to determine whether one of the types is uint, if the contract contains an arithmetic operation. (2) MSmart determines if the contract makes use of the SafeMath standard library. (3) The presence or absence of an "if" or "require" judgment statement before an arithmetic operation occurs. MSmart will identify an integer overflow vulnerability if there is an arithmetic computation that does not utilize the SafeMath library and an "if" or "require" judgment statement.

### 4.2. Timestamp Dependence

When mining a block, miners must set a block timestamp for the block. Typically, the block timestamp is set to the current time on the miner's local system. However, miners can change this time by roughly 900 s, and other miners will still accept this block [37]. Specifically, miners verify whether the block timestamp is larger than the block timestamp of the preceding block and within 900 s of the block timestamp on their local system after receiving a new block and doing additional validity checks. Therefore, attackers may alter the execution outcomes of timestamp-dependent contracts by selecting alternate block timestamps.

The timestamp dependence detection algorithm is shown in Algorithm 2.

---

**Algorithm 2:** The timestamp-dependent algorithm.

---

**Input:** a smart contract P
**Output:** timestamp dependence vulnerabilities
1: Convert the source code to the corresponding path diagram
2: Match according to the path expression pattern
3: **if** pattern matching found timestamp (such as "now") **then**
4:　　　**if** the timestamp is assigned a value and then an arithmetic operation occurs on the value
　　　　　　**then**
5:　　　　　**return** timestamp dependence vulnerability
6:　　　**else**
7:　　　　　**return** 0
8:　　　**end if**
9: **else**
10:　　　**return** 0
11: **end if**
12: Integrate all discovered vulnerabilities

---

In the timestamp dependence detection algorithm, MSmart mainly checks the contracts as follows: (1) MSmart checks whether the block timestamp appears in the contract first. (2) It is determined if a timestamp has been assigned to the value before determining whether an arithmetic operation is carried out on this value. If all of the above exist, MSmart has identified a vulnerability.

### 4.3. Self-Destruct

Experimental analysis: The self-destruct function can force the balance in the account to be sent to a specified address. The attacker will steal the ether in the contract if the address of the input account differs from the address of the contract owner. Therefore, it is important to confirm that the sending address is the address of the contract owner.

Based on the above analysis, the basic ideas for improvement are shown in Algorithm 3:

The self-destruct vulnerability detection algorithm is shown in Algorithm 3.

---

**Algorithm 3:** The self-destruct vulnerability detection algorithm.

---

**Input:** a smart contract P
**Output:** Vulnerability
1: Convert the source code to the corresponding path diagram
2: Match according to the path expression pattern
3: **if** pattern matching found keyword "suicide" or "self-destruct" **then**
4:　　　**if** there is no contract owner judgment or subsequent address are not contract owners
　　　　　　**then**
5:　　　　　**return** self-destruct vulnerability
6:　　　**else**
7:　　　　　**return** 0
8:　　　**end if**
9: **else**
10:　　　**return** 0
11: **end if**
12: Integrate all discovered vulnerabilities

---

In the self-destruct detection algorithm, we mainly check the contracts as follows: (1) MSmart finds whether there is a self-destruct function in the contract. (2) When the self-destruct function transfers ether, MSmart verifies that the address is the address of the contract owner. (3) Nowadays, the self-destruct function "suicide" is less used, and it is necessary to add another "selfdestruct" function to the judgment. If the destination address of the self-destruct function is not the owner of the contract, MSmart determines that a vulnerability exists.

### 4.4. Delegatecall

If there is "*msg.sender*" or "*msg.value*" after the delegatecall, there is a significant security risk. To avoid this vulnerability is to not use delegatecall functions.

Based on the above analysis, the basic ideas for improvement are shown in Algorithm 4: The delegatecall detection algorithm is shown in Algorithm 4.

---

**Algorithm 4:** delegatecall detection algorithm.

---

**Input:** a smart contract P
**Output:** delegatecall vulnerabilities
1: Convert the source code to the corresponding path diagram
2: Match according to the path expression pattern
3: **if** pattern matching found keyword "delegatecall" **then**
4:      **if** the variable msg.sender or msg.value exists after the keyword "delegatecall" **then**
5:          **return** delegatecall vulnerability
6:      **else**
7:          **return** 0
8:      **end if**
9: **else**
10:      **return** 0
11: **end if**
12: Integrate all discovered vulnerabilities

---

In the delegatecall algorithm, we mainly check the contracts as follows: (1) MSmart checks the contract for the presence of "delegatecall". (2) If "delegatecall" is present, MSmart determines whether "*msg.value*" or "*msg.data*" is present after delegatecall. If all of the above exist, MSmart has identified a vulnerability.

### 4.5. Denial of Service

The "require" judgment must be met for the contract to continue, as shown by the analysis above; hence, a possibility must be taken into account. The attacker will finally attack successfully if the "require" judgment statement keeps failing and producing an anomalous outcome. Therefore, the attention should be paid to the presence of the transfer function in the "require" judgment statement.

Based on the above analysis, the basic ideas for improvement are shown in Algorithm 5: The DOS detection algorithm is shown in Algorithm 5.

---

**Algorithm 5:** DOS detection algorithm.

---

**Input:** a smart contract P
**Output:** DOS vulnerabilities
1: Convert the contract P into an intermediate representation (IR)
2: Match according to the path expression pattern
3: **if** pattern matching found conditional statement (such as "require") **then**
4:      **if** there are transfer keywords (such as "send") within the determine statements **then**
5:          **return** DOS vulnerability
6:      **else**
7:          **return** 0
8:      **end if**
9: **else**
10:      **return** 0
11: **end if**
12: Integrate all discovered vulnerabilities

---

In the DOS algorithm, we mainly check the contracts as follows: (1) MSmart first checks if there is a "require" statement in the contract. (2) MSmart determines whether

a transfer function exists in the statement (such as "call"). If both take place, a DOS vulnerability is identified.

## 5. Experiment

We implemented an improved tool called MSmart, tested a large number of smart contracts, and thoroughly analyzed several common contracts in order to verify the efficacy of the detection algorithm proposed. This proves that the improved algorithm has better performance. Through trials, we timed how long it took each tool to find vulnerabilities in 100 and 1000 smart contracts and compared the findings. This proves that it takes less time for MSmart to detect the same vulnerabilities.

### 5.1. MSmart vs. Smartcheck on Large Data Sets

As of August 2022, the source codes of 6000 contracts downloaded from Etherscan were filtered and verified [38], and MSmart runs on this data set. MSmart has lowered the number of false negatives and false positives when compared to the outcomes of Smartcheck. Table 2 displays the data after the same outcomes were eliminated:

**Table 2.** Comparison of MSmart and Smartcheck on large data sets.

| Smart Contract Vulnerability Categories | Number of Smartcheck Vulnerability Analysis | Smart Contract Vulnerability Categories | Number of MSmart Vulnerability Analysis |
|---|---|---|---|
| integer overflow | 0 | integer overflow | 2115 |
| timestamp dependence | 44 | timestamp dependence | 1175 |
| DOS | 0 | DOS | 1102 |
| self-destruct | 330 | self-destruct | 154 |
| delegatecall | 0 | delegatecall | 7 |

From Table 2, we can find that:

(1) Integer overflow vulnerability: Smartcheck did not protect against this vulnerability, and MSmart reported 2115 vulnerabilities. A total of 72% of the integer overflow vulnerabilities were discovered to be true positives after manual verification. Because the protection against integer overflows is building a function, it is difficult to implement unified protection because the function name is not fixed. This is the main reason why contracts reporting integer overflow vulnerabilities greater than 4 frequently reported false positives.

(2) Timestamp dependence: Smartcheck reported 44 vulnerabilities, and MSmart reported 1175 vulnerabilities. A total of 80% of the timestamp dependence vulnerabilities were discovered to be true positives after manual verification. It is risky to use the block timestamp as a pseudo-random number, if it is used as a trigger condition to carry out certain significant operations, even if it is altered, the result will not be affected, and MSmart still identifies it as a vulnerability.

It is mentioned in Section 4.2 when miners can obtain results in their favor by freely manipulating the block timestamps. Therefore, there is a high risk of timestamp-dependent vulnerabilities. In the timestamp-dependent vulnerability detection, MSmart detected 1175 vulnerabilities in total, and Smartcheck detected only 44 vulnerabilities. For the relatively large difference between the results of the two tools, we randomly selected six contracts from the vulnerability detection reports for validation, and the final validation results are shown in Table 3.

**Table 3.** Timestamp issue statistics results.

| Contract | MSmart | Smartcheck |
|---|:---:|:---:|
| Governmental_survey.sol | ✓ | ✗ |
| Lottopollo.sol | ✓ | ✗ |
| Roulette.sol | ✓ | ✗ |
| Time_crowdsale.sol | ✓ | ✗ |
| Time.sol | ✓ | ✗ |
| TimeFame.sol | ✓ | ✗ |

(3) DOS: SmartCheck did not protect against this vulnerability and MSmart reported 1102 vulnerabilities. Analysis of this issue is more challenging since MSmart is a static smart contract analysis tool. As a result, there are some false positives in detecting this vulnerability. MSmart will notify users if a smart contract has this vulnerability.

(4) Self-destruct: Smartcheck reported 330 vulnerabilities, and MSmart reported 154 vulnerabilities. In the manual verification of detected self-destruct vulnerabilities, it was discovered that this vulnerability had a low false positive rate. The self-destruct vulnerability's fixed format is the primary cause. MSmart is thus more effective at guarding against this vulnerability.

(5) Delegatecall: SmartCheck did not protect against this vulnerability and MSmart reported seven vulnerabilities. These seven vulnerabilities are false positives, but if the situation changes, they become real vulnerabilities.

It is clear from the data above that MSmart can effectively analyze timestamp and integer overflow vulnerabilities. Due to the high mistake frequency of timestamp dependencevulnerabilities and integer overflow vulnerabilities, this enhancement may provide more effective protection. MSmart can also analyze DOS and delegatecall vulnerabilities. If any of these vulnerabilities emerge, they may be reported. As a result, MSmart's vulnerability protection is more effective than Smartcheck's.

*5.2. MSmart vs. Smartcheck on Specific Contracts*

All vulnerabilities discovered by the tool are manually marked as "true positives" (TP) or "false positives" (FP) (TP is the actual number of vulnerabilities, FP is the number of false positives). For each detection tool, the false discovery rate (FDR) was defined as the number of vulnerabilities for that tool divided by the number of all vulnerabilities reported by that tool. The false negative rate (FRN) is the number of FNs for the tool divided by the number of all true discoveries (by any tool or manually), which is the sum of the tool's TP and FN (FN is the number of false negatives. The calculation formula is as follows:

$$FDR = FP / (TP + FP) \tag{1}$$

$$FNR = FN / (TP + FN) \tag{2}$$

Due to the fact that Smartcheck does not support delegatecall vulnerabilities, DOS vulnerabilities, and integer overflow vulnerabilities. Therefore, timestamp-dependent vulnerabilities and self-destruct vulnerabilities are compared here. There are 20 smart contracts in Project1, 10 of which are timestamp dependence vulnerabilities, and 10 of which appear to be timestamp-dependently vulnerable but are not. A total of 20 smart contracts make up project2, 8 of which have self-destruct vulnerabilities, and 12 of which appear to have self-destruct vulnerabilities but have no vulnerabilities. The analysis results are shown in Table 4.

Table 4 shows that the FNR of these three smart contracts has decreased from 83.3%, and 75.0% to 50.0% and 25.0%, respectively, and that the FDR has increased from 9.1%, and 20.0% to 33.3% and 42.8%. Since the FNR has decreased more noticeably, MSmart has likely been successful in identifying contract vulnerabilities and reducing false negative. FDR has been risen compared to Smartcheck, which shows the rise of MSmart's accuracy rate.

**Table 4.** Comparison of MSmart and Smartcheck on specific projects.

| Project | Indicator | Smartcheck | MSmart |
|---------|-----------|------------|--------|
| | *TP/FP/FN* | 1/10/5 | 5/10/5 |
| Project1 | *FDR* (%) | 9.1 | 33.3 |
| | *FNR* (%) | 83.3 | 50.0 |
| | *TP/FP/FN* | 2/8/6 | 6/8/2 |
| Project2 | *FDR* (%) | 20.0 | 42.8 |
| | *FNR* (%) | 75.0 | 25.0 |

In conclusion, MSmart has significantly decreased the false negative rate when compared to Smartcheck. Considering the security of smart contracts, if the detection tool has false negatives, it is fatal to the security of smart contracts. Naturally, it cannot reach 100% detection due to a flaw in its detection technique, so some false positives are unavoidable.

*5.3. MSmart vs. Other Tools*

Oyente is the first smart contract detection tool that uses contract control flow diagrams and symbolic execution to find smart vulnerabilities. Transaction order dependencies, integer overflow vulnerabilities, etc. are all detected by Oyente. Oyente has high false negatives but few false positives. The official suggested tool for detecting smart contracts in Ethereum is Mythril, which uses symbolic execution to find smart contract vulnerabilities. The majority of vulnerabilities can be successfully detected via Mythril. Slither is a smart contract vulnerability detection tool that uses an intermediate representation and can identify vulnerabilities from a syntactic standpoint. Therefore, we choose these tools. Next, the test dataset is the dataset given by smartbugs [39] and a small number of validated vulnerable smart contracts in order to make the experiment realistic and fair. Finally, through analysis results, MSmart has made a series of improvements in both false negatives and false positives. The analysis results are shown in Table 5.

**Table 5.** Comparison of MSmart and other tools.

| Smart Contract Vulnerability Categories (Number of Vulnerabilities) | Oyente | Mythril | Slither | Smartcheck | MSmart |
|---|---|---|---|---|---|
| integer overflow (15) | 11 | 12 | 0 | 0 | 13 |
| timestamp dependence (5) | 1 | 0 | 1 | 1 | 2 |
| DOS (6) | 0 | 0 | 0 | 0 | 2 |
| self-destruct (5) | 0 | 5 | 5 | 1 | 5 |
| Delegatecall (5) | 0 | 5 | 5 | 0 | 5 |

Note: the numbers in the table all represent the number of vulnerabilities detected by each tool.

The experimental analysis shows the following:

(1) For integer overflow vulnerabilities: Oyente, Mythril and MSmart can detect most of the integer overflow vulnerabilities. On the other hand, this kind of smart contract vulnerability detection is not supported by Slither and Smartcheck. MSmart detects 13 out of 15 vulnerable smart contracts (note: MSmart has more detected vulnerabilities than other tools). This demonstrates that MSmart is superior to other mitigation techniques for integer overflow vulnerabilities.

(2) For timestamp dependence vulnerabilities: Oyente, Slither, Smartcheck, and MSmart can detect this type of vulnerability. This kind of smart contract vulnerability detection is not supported by Mythril. Oyente detects if a money flow depends on the timestamp. Therefore, some of the timestamp vulnerabilities that are not related to money flow cannot be detected. Smartcheck and Slither's detection algorithms are not flawless. MSmart offers a more thorough defense against timestamp dependence vulnerabilities.

(3) For DOS vulnerabilities: MSmart can detect this type of vulnerability. On the other hand, this kind of smart contract vulnerability detection is not supported by Oyente,

Mythril, Slither, and Smartcheck. Based on the analysis's findings, MSmart has been able to guard against this kind of vulnerability.

(4) For self-destruct vulnerabilities: Mythril, Slither, Smartcheck, and MSmart can detect this type of vulnerability. This kind of smart contract vulnerability detection is not supported by Oyente. Smartcheck's detection rules for such vulnerabilities are not perfect. Therefore, only one can be detected. Mythril, Slither, and Smartcheck can detect all of them.

(5) For delegatecall vulnerabilities: Mythril, Slither and Smartcheck can detect this type of vulnerability. This kind of smart contract vulnerability detection is not supported by Oyente and Smartcheck.

In conclusion, MSmart effectively protects against the aforementioned five categories of vulnerabilities. MSmart cannot analyze all the vulnerabilities in time, but it has been able to defend most of them.

### 5.4. Time Efficiency Comparison and Analysis

Along with false positive and false positive rates, the time efficiency in smart contract detection is also crucial for detecting smart contract vulnerabilities. Testing the time consumption of Oyente, Slither, and MSmart allowed for the analysis of the time efficiency of MSmart, as shown in Table 6.

**Table 6.** Comparison of MSmart and other tools for time efficiency.

| Smart Contract Detection Tool | Time Consumption/s | |
|:---:|:---:|:---:|
| | 100 Contracts | 1000 Contracts |
| Oyente | 21,045.3 | 223,056.7 |
| Slither | 54.0 | 567.3 |
| Smartcheck | 36.4 | 400.1 |
| MSmart | 10.3 | 80.0 |

The experiment defines a variable $R_t$ representing the ratio, which is used to denote the ratio of the time consumed by each tool and MSmart in the case of executing the same number of contracts. The calculation formula is as follows:

$$R_{t(tool)} = T_{(tool)} / T_{(MSmart)} \tag{3}$$

The results are shown in Table 7.

**Table 7.** The value of $R_{t(tool)}$.

| Tool | Oyente | Slither | Smartcheck |
|:---:|:---:|:---:|:---:|
| $R_{t(tool)}$ (100 contracts) | 2043.2 | 5.2 | 3.5 |
| $R_{t(tool)}$ (1000 contracts) | 2788.2 | 7.1 | 5.0 |

Table 7 shows that the value of $R_{t(Oyente)}$, $R_{t(Slither)}$, $R_{t(Smartcheck)}$ decreases sequentially (note: $R_{t(Smartcheck)} > R_{t(MSmart)} = 1$). To facilitate MSmart batch detection for smart contracts, we used six threads to implement parallel batch processing between each Solidity file, so MSmart is faster than Smartcheck. Oyente leverages symbolic execution to detect vulnerabilities. By translating the source code into bytecode and then detecting smart contract vulnerabilities based on the contract flowchart. Due to the use of constraint solver, the time consumption is much higher than MSmart. Slither not only detects smart contract vulnerabilities but also analyzes the syntax. Therefore, Slither takes slightly longer to detect than MSmart and Smartcheck. These show that MSmart has substantially greater analytical efficiency than Oyente Slither and Smartcheck, and that their ratio rises as the number of contracts increases.

### 6. Conclusions

We investigate the fundamentals of vulnerabilities such as integer overflow and, based on the research, create a new smart contract detection tool called MSmart. When defending against particular vulnerabilities, MSmart has shown to be more effective. MSmart has good protection against timestamp vulnerabilities and integer overflow vulnerabilities, according to a comparison of MSmart with other tools. MSmart can also report these kinds of vulnerabilities whenever they occur, such as DOS vulnerabilities, delegatecall vulnerabilities, and self-destruct vulnerabilities.

MSmart mostly depends on the vulnerability criteria developed for analysis, yet attackers are able to go around the defense and launch an assault. It is challenging for MSmart to rely only on current rules in order to achieve more accurate protection. Therefore, dynamic execution and static analysis may be integrated in future studies. For instance, it is possible to completely fulfill the static analysis of integer overflow vulnerabilities and to use dynamic execution to confirm the delegatecall vulnerabilities.

**Author Contributions:** Conceptualization, J.F., X.Z.; investigation and methodology, J.F., X.Z.; writing of the original draft, J.F.; writing of the review and editing, J.F., X.Z.; validation, X.C.; software, J.F.; data curation, J.F., X.Z. All authors have read and agreed to the published version of the manuscript.

**Funding:** The National Natural Science Foundation of China: (61972360, 62072392).

**Institutional Review Board Statement:** Not applicable.

**Informed Consent Statement:** Not applicable.

**Data Availability Statement:** Not applicable.

**Conflicts of Interest:** The authors declare no conflict of interest.

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
