# Peer review of "MSmart: Smart Contract Vulnerability Analysis and Improved Strategies Based on Smartcheck"

_applsci, doi:10.3390/app13031733_

Round 1

Reviewer 1 Report

This paper was a very interesting topic and easy to read. Due to the number of cryptocurrecy hacks, this area of research is very exciting at the moment. I believe this paper would be suitable after a significant revision; in its current form, it reads as though it has been excerpted from a larger writing and is missing a portion of the supporting material. Some recommended revisions are: 

1. Add an inventory and more extensive background for recently occurring hacks. The authors reference that hacks are occurring, but it is important to know how often each vulnerability is being exploited and what is the larger pattern among projects that are vulnerable. Without this, it is hard to know that the experiment solves the problem. Are a majority of hacks happening within the categories that MSmart performs better at? If not, it is difficult to prove the findings as significant. 

2. The numbering of sections and figures is not consistent. Examples of this are at lines 140, 197, and 263. It is recommended to update all header, figure, and table references. 

3. The keywords that are used to detect items such as "self-destruct" or "suicide" are quite a narrow match, considering that the same function could carry other names. Is this a reliable filter? It is recommended to provide an expanded explanation of why other analysis tools do not use such narrow matches. If they do, it is recommended to explain why they are not matching the same terms as MSmart. 

4. To expand on the prior point, in table 1, there is a comparison of performance. Please provide more context for why there is such as large gap in performance. Why are the methods used for MSmart not adopted in other tools. Please provide an expanded explanation for this for each of the 5 categories tested and included supporting citations for the reliability and performance claims in each case.

5. Table 3 is using quite a narrow sample. Even through the sample is random, using only 3 cannot be considered representative. It is recommended to revise this analysis so that the sample is larger, more diverse, or otherwise proven as scientifically representative. 

6. At line 362 the authors reference symbolic execution and intermediate representation. Please expand this note to explain the differences between the two, why tools would choose one over the other, and the implications this has for MSmart. 

7. In section 5.3 please provide more information as to why competing tools fail at certain detections. Are they intentionally making a tradeoff against other results? Are certain tests less reliable? Are certain exploits less common? I am unsure if the difference in detections is of practical importance without this context. 

8. In section 5.4 the performance values provided can be considered relative without additional context. Please expand this section to include information on the agorithmic complexity of each of the tests that MSmart runs, and a direct comparison against tests run by competing systems. In current form, I could simply assume that competing products are more thorough during tests or doing multiple variation of each scan pass before finalizing a result. An underlying analysis of why there is a performance gap is needed to place the claims into a provable context.

Reviewer 2 Report

This paper is about an improved version of an existing framework (Smartcheck)) by introducing middleware (intermediate representation rules) for smart contract vulnerability detection. The comparative performance evaluation was done based on five smart contract attacks: integer overflow, timestamp, self-destruct function, proxy call, and DoS.

Here are the strengths of the paper: providing good background of the five vulnerabilities helping new readers to comprehend, and having impressive performance improvement.

Here are the suggested improvements: summarising the major differences between MSmart and Smartcheck in a table for the benefits of readers, and correcting some subsection numbering errors in Section 2.

Reviewer 3 Report

I believe that this paper will be of interest to the readership of your journal.

 Below are several suggestions that I hope will be helpful in the paper:

Abstract fully and clearly expresses goal, but the main methods and the methodology needs to be explained.

Referencing could be more accurate and unified – sometimes whole name is written, sometimes only the surname.

Section heading 2 and section sub heading 3.4, 3.5, 3.6 (p.4-5) are misplaced

In line 157 (p.5) and line 169 is duplicate heading (.2.2)
